# Melatonin and MitoEbselen-2 Are Radioprotective Agents to Mitochondria

**DOI:** 10.3390/genes14010045

**Published:** 2022-12-23

**Authors:** Tsutomu Shimura, Rina Shiga, Megumi Sasatani, Kenji Kamiya, Akira Ushiyama

**Affiliations:** 1Department of Environmental Health, National Institute of Public Health, Wako 351-0197, Japan; 2Meiji Pharmaceutical University, Kiyose 204-8588, Japan; 3Department of Experimental Oncology, Research Center for Radiation Genome Medicine, Research Institute for Radiation Biology and Medicine (RIRBM), Hiroshima University, Hiroshima 734-8553, Japan

**Keywords:** glutathione peroxidase, melatonin, mitochondria, MitoEbselen-2, radiation protection

## Abstract

Mitochondria are responsible for controlling cell death during the early stages of radiation exposure, but their perturbations are associated with late effects of radiation-related carcinogenesis. Therefore, it is important to protect mitochondria to mitigate the harmful effects of radiation throughout life. The glutathione peroxidase (GPx) enzyme is essential for the maintenance of mitochondrial-derived reactive oxygen species (ROS) levels. However, radiation inactivates the GPx, resulting in metabolic oxidative stress and prolonged cell injury in irradiated normal human fibroblasts. Here, we used the GPx activator *N*-acetyl-5-methoxy-tryptamine (melatonin) and a mitochondria-targeted mimic of GPx MitoEbselen-2 to stimulate the GPx. A commercial GPx activity assay kit was used to measure the GPx activity. ROS levels were determined by using some ROS indicators. Protein expression associated with the response of mitochondria to radiation was assessed using immunostaining. Concurrent pre-administration or post-administration of melatonin or MitoEbselen-2 with radiation maintained GPx activity and ROS levels and suppressed mitochondrial radiation responses associated with cellular damage and radiation-related carcinogenesis. In conclusion, melatonin and MitoEbselen-2 prevented radiation-induced mitochondrial injury and metabolic oxidative stress by targeting mitochondria. These drugs have the potential to protect against acute radiation injury and late effects of carcinogenesis in a variety of radiation scenarios assuming pre-administration or post-administration.

## 1. Introduction

Humans exposed to high levels of radiation lose function of tissues and organs due to cell death, and clinical symptoms, such as hematopoietic subsyndrome and gastrointestinal subsyndrome, manifest as acute radiation syndrome with an incubation period of several weeks or less [1]. Radiation protection is critical for improving care for radiation victims in a radiation emergency. In addition to acute radiation effects, late radiation effects include carcinogenesis that manifests decades later [2]. Various radiation-protective agents, such as antioxidants, nutrients, and phytochemicals, have been identified for use in a variety of radiation scenarios [3]. In humans, potassium iodide has been shown to block internal thyroid exposure to I-131 radioiodine [4]. Patt et al. reported that administering cysteine to animals prior to irradiation increased survival after total body irradiation (TBI) by scavenging free radicals and neutralizing the effect of radiation [5]. Granulocyte colony-stimulating factor (G-CSF) administration promotes the proliferation and differentiation of bone marrow progenitor cell populations and significantly increases mouse survival after TBI [6]. Traditional radioprotective agent research has primarily focused on pre-administration for radiation workers or cancer patients who are likely to be exposed to radiation [7]. On the other hand, radiation accidents are unintended or unexpected exposures; therefore, the development of radiation-protective agents with the effect of promoting recovery from radiation damage after exposure is an urgent issue. Despite advances in basic research, radiation-protective agents have yet to be clinically tested due to serious side effects [3]. Because there are still difficult problems to solve for translating agents from animal testing to practical implementation, more research is needed to understand the mechanisms of protective action for radiation-protective agents.

Mitochondria regulate cell death by releasing cytochrome C, which acts as a sensor of cell damage [8]. Mitochondria also play a role in radiation-induced oxidative stress responses in both acute radiation injury and late-onset carcinogenic effects [9,10]. We have previously reported that radiation activates ataxia-telangiectasia mutated (ATM)-mediated mitochondrial respiration, which is required for energy supply to respond to nuclear DNA damage [11,12]. Stimulating mitochondrial respiration increases the generation of delayed reactive oxygen species (ROS), which in turn mediates oxidative stress as an indirect effect of radiation [12]. Thus, mitochondria generate ROS and are vulnerable to ROS-mediated oxidative stress. Mutations in mitochondrial DNA have been found in a variety of human cancers [13,14]. We further reported that mitochondrial ROS activate transforming growth factor-β (TGF-β) signaling, resulting in tumor microenvironment formation in radiation-related carcinogenesis [15]. Thus, mitochondrial oxidative stress is associated with carcinogenesis. Aside from nuclei, mitochondria are believed to be an important radiation target for radiation protection.

Glutathione (GSH) is an antioxidant that maintains intracellular redox homeostasis. Mitochondrial GSH is an important molecule in mitochondrial oxidative stress control. GSH detoxifies ROS by converting them to GSH disulfide (GSSG) and H_2_O via the enzyme activity of GSH peroxidase (GPx) [16,17,18]. NADPH-dependent GSH reductase then converts GSSG back to GSH. We recently reported that radiation has been shown to inactivate GPx, which increases ROS levels and causes ROS-mediated oxidative stress in human fibroblasts [19]. Vitamin C and the flavonoid epicatechin found in tea leaves have the potential to protect mitochondria from radiation [20]. In this paper, we challenge the development of a new radioprotective agent that targets mitochondria. We focused on GPx-related agents, such as melatonin (N-acetyl-5-methoxy-tryptamine) and MitoEsbselen-2, to develop radiation protective agents with fewer side effects. Melatonin is present in almost all organisms to regulate the antioxidant system and circadian rhythms [21,22,23,24]. Melatonin is used clinically for sleep disorders of circadian etiology and neurological degenerative diseases [25,26,27]. Melatonin has been shown to stimulate GPx and act as a free radical scavenger [28,29]. MitoEbselen-2, a mitochondrial-targeted glutathione peroxidase mimic, is a radiation mitigator [30].

## 2. Materials and Methods

### 2.1. Cell Culture, Drugs, and Irradiation

TIG-3 and MRC-5 normal human diploid lung fibroblasts were obtained from the Health Science Research Resources Bank and Riken Cell Bank, respectively, and cultured in α minimum essential medium (Nacalai Tesque, Kyoto, Japan) containing 10% heat-inactivated fetal calf serum. The cells were treated with 0.1 μM melatonin (FUJIFILM Wako Pure Chemical Co., Osaka, Japan) or 20 μM MitoEbselen-2 (MedKoo Biosciences, Inc., Chapel Hill, NC, USA) 2 h before (pre-radiation treatment) or 2 h after (post-radiation treatment) ionizing radiation (IR). A 150-kVp X-ray generator (Model MBR-1505R2, Hitachi-Medico Co., Hitachi, Tokyo, Japan) equipped with 0.5 mm Cu and 0.1 mm Al filters was used for irradiation.

### 2.2. GPx Assay

Cell extracts were taken 1 day after irradiation. A GPx Activity Assay Kit (Biovision Inc., Mountain View, CA, USA) was used to measure the GPx activity in the indicated samples. The activities of GPx were determined by evaluating the reduction of nicotinamide adenine dinucleotide phosphate (NADP+) to nicotinamide adenine dinucleotide phosphate (NADPH) in the cell lysates following the instruction from the manufacturer’s protocols. NADPH was determined by measuring spectrophotometric absorbance at 340 nm. Protein content in the enzymatic extracts was determined by the Bradford protein assay. The activity was expressed as μmol NADPH/min per mg protein.

### 2.3. ROS Detection

The redox-sensitive 2′,7′-dichlorofluorescin diacetate (DCFDA; Sigma Genosys, The Woodlands, TX, USA) at 5 μM for 30 min was used to measure intracellular ROS levels. The DCFDA does not differentiate between different types of ROS [31]. Mitochondria-delivered ROS was measured with 2.5 μM MitoSOX-red for 10 min (Thermo Fisher Scientific, Inc., Waltham, MA, USA). OxiORANGE dye at 1 μM for 20 min (Goryo Chemical Inc., Sapporo, Japan) detects hydroxyl radicals (•OH). A FACScan automated flow cytometer (Becton, Dickinson and Company, Franklin Lakes, NJ, USA) was used to quantify stained cells. Data were normalized to non-irradiated controls.

### 2.4. Cell Counting

TIG-3 and MRC-5 cells (1.0 × 10^5^ cells) were seeded into 25 cm^2^ flasks (Thermo Fisher Scientific). The cells were incubated overnight, X-rayed the next day, and then further incubated for 3 days. Cells were trypsinized and total number of cells was determined by using a hemocytometer and an optical microscope.

### 2.5. Immunofluorescence

The staining protocol has previously been described [32]. The cells were fixed with 4% paraformaldehyde 1 day after IR. Antibodies were listed as follows: phosphorylated AMP-activated protein kinase (AMPK, PA5-37821, Invitrogen, Carlsbad, CA, USA), γ-H2A histone family member X (γ-H2AX, 05-636, Millipore, Billerica, MA, USA), Nuclear factor erythroid 2-related factor 2 (Nrf2) (ab31163, Abcam, Cambridge, MA, USA), Parkin (14060-1-AP, Proteintech, Wuhan, China), α-smooth muscle actin (α-SMA) (A2547, Sigma), translocase of outer membrane 20 (TOM20, 612278, BD, Biosciences, San Jose, CA, USA), secondary antibodies conjugated to Alexa Fluor 488 (A11034, Molecular Probes, Eugene, OR, USA) or Alexa Fluor 647 (A21236, Molecular Probes). The images were acquired and analyzed using a Keyence BZ-X700 fluorescence microscope and Hybrid Cell Count Software (BZ-II Analyzer, Keyence Corporation, Oosaka, Japan). More than 50 cells were manually counted for each data point.

### 2.6. Statistical Analysis

The data represent the mean ± standard deviation and were obtained from at least three independent samples. Following one-way ANOVA, Dunnett’s tests were used to detect significant differences between the means of three or more independent groups. Student’s *t*-tests were used to compare two groups.

## 3. Results

### 3.1. Maintaining GPx Activity following Radiation Exposure by Pre-Radiation or Post-Radiation Treatment with Melatonin or MitoEbselen-2

GPx enzymes play an important role in scavenging ROS with GSH [17,18]. However, we previously reported that radiation inactivates GPx activity in a dose-dependent manner, causing ROS control to be disrupted in TIG-3 and MRC-5 cells [19]. The irradiated cells were treated with GPx activator melatonin or a GPx mimic MitoEbselen-2 to reduce the effect of radiation on GPx activity. The effects of these drugs were evaluated by measuring GPx activity in the indicated samples (Figure 1). Melatonin treatment either before or after radiation prevented radiation-induced inactivation of GPx activity in both cells (Figure 1A,B). Similarly, radiation had no effect on GPx activity when cells were pre-treated and post-treated with MitoEbselen-2 (Figure 1A,B).

### 3.2. Scavenging ROS by GSH-Related Drugs

The effect of melatonin and MitoEbselen-2 on intracellular redox state was examined by measuring ROS levels in TIG-3 and MRC-5 cells using some ROS indicators. The DCFDA probe indicates the oxidative stress conditions. Radiation induced perturbation in redox control, as evidenced by increased DCFDA staining in TIG-3 and MRC-5 cells (Figure 2A). However, pre-radiation and post-radiation melatonin or MitoEbselen-2 treatment suppressed the increase in DCFDA staining in both irradiated cells (Figure 2A). The activation of mitochondrial oxidative phosphorylation in response to DNA damage causes radiation-induced delayed ROS generation [11]. We previously reported that GSH-mediated redox homeostasis maintains mitochondrial ROS levels below 2 Gy, whereas high ROS levels remained at 24 h after exposure to high doses (>5 Gy; Figure 2B) [19,33]. Melatonin or MitoEbselen-2 pre-radiation and post-radiation treatment had no effect on mitochondrial ROS generation after radiation (Figure 2B). Hydroxyl radicals (•OH) cause the most severe cell damage of any ROS [34]. •OH and hypochlorous acid (HClO) can be detected selectively in live cells using the dye (Goryo Chemical; [35]). Radiation causes an increase in •OH levels, whereas melatonin or MitoEbselen-2 inhibited the induction of an increase in •OH level after irradiation in TIG-3 and MRC-5 cells (Figure 2C).

### 3.3. Mitigating Radiation Effects on Cell Growth Suppression by Treatment with Melatonin or MitoEbselen-2

The number of cells in TIG-3 cells was counted 3 days after irradiation to evaluate the protective role of melatonin and MitoEbselen-2 on cell growth (Figure 3A). When cells were exposed to radiation doses >2.5 Gy, their growth was inhibited in a dose-dependent manner. In contrast, melatonin-treated cells showed growth inhibition at >5 Gy, but MitoEbselen-2-treated cells showed no growth retardation. Thus, combining radiation with melatonin or MitoEbselen2 reduced the effect of radiation on cell growth. Similar results were obtained in MRC-5 cells (Appendix A). Radiation-induced DNA double strand breaks were examined by using a marker γ-H2AX. γ-H2AX foci were found 24 h after 10-Gy irradiation (Figure 3B). In contrast, pre-radiation and post-radiation melatonin or MitoEbselen-2 treatment did not significantly increase γ-H2AX fluorescence levels (Figure 3C). We further examined the Nrf2-mediated antioxidant response. Radiation alone activated Nrf2, as evidenced by nuclear accumulation of Nrf2 in TIG-3 cells (Figure 3B). Radiation with pre-radiation and post-radiation melatonin or MitoEbselen-2 treatment did not induce such Nrf2 radiation responses (Figure 3D).

### 3.4. Effects of Melatonin and MitoEbselen-2 on Mitochondrial Radiation Response

Mitochondrial autophagy (mitophagy) maintains mitochondrial quality through degradation [36]. Parkin recognizes dysfunctional mitochondria with low membrane potential [37]. Radiation with 10 Gy revealed Parkin staining in Tom20-stained mitochondria as a green color (Figure 4A). In contrast, Parkin foci were not detected by radiation when cells were pre-treated and post-treated with melatonin or MitoEbselen-2 (Figure 4B). Tom20 is used to determine the change in mitochondrial amounts. The fluorescence intensity values of Tom20 staining in TIG-3 cells increased with radiation exposure. However, pre-radiation and post-radiation treatment with melatonin or MitoEbselen-2 had no effect on the Tom20 radioresponse (Figure 4C).

### 3.5. Fibroblast Activation

We next investigated whether melatonin and MitoEbselen-2 could prevent the appearance of activated fibroblasts, which were identified as cells with α-SMA-positive fibers (Figure 5A). We previously reported that radiation induced increase in α-SMA protein expression in human fibroblasts [38]. Immunostaining revealed that activated fibroblasts showed altered morphology (flatter and larger) with a fiber-like staining pattern of α-SMA as reported previously [15] (Figure 5A). Ten-Gy radiation induced α-SMA-positive cells in TIG-3 and MRC-5 cells. Melatonin and MitoEbselen-2 pre-radiation (Figure 5B) and post-radiation treatments (Figure 5C) inhibited the induction of α-SMA-positive cells after radiation.

## 4. Discussion

People’s anxiety about radiation has become a heavy burden both physically and mentally in the aftermath of the Fukushima disaster, increasing health risks regardless of exposure dose [39]. Studies on the effect of radiation on mitochondrial oxidative stress can help us to understand the mechanism of radiation carcinogenesis. The radiation risk knowledge gained will be used to disseminate basic radiation knowledge. The research findings are important not only for laying the groundwork for a new radiation protection system but also for deepening the general public’s understanding of radiation.

Mitochondria are the sites of intracellular ROS generation during the mitochondrial energy supply process [40]. The GPx is involved in the maintenance of mitochondrial-derived ROS levels. However, radiation perturbs GSH-mediated redox control by inactivating the GPx, causing metabolic oxidative stress and prolonged cell injury due to delayed mitochondrial ROS generation [10,19]. We here demonstrated that pre-radiation and post-radiation treatment with melatonin or MitoEbselen-2 preserved GPx activity after irradiation. The effects of melatonin or MitoEbselen-2 on GPx-mediated redox control were depicted in Figure 6. The use of DCFDA staining revealed that combining melatonin or MitoEbselen-2 with radiation did not perturb cellular redox homeostasis in human fibroblasts. Both drugs had no effect on the generation of mitochondrial ROS as measured by MitoSOX-red staining, but they did suppress the increase in radiation-induced •OH levels. The absence of γ-H2AX foci induction and nuclear Nrf2 accumulation indicated that these drugs eliminated radiation-induced DNA damage and antioxidant responses. Furthermore, mitochondrial damage was tracked using the Parkin antibody. Melatonin and MitoEbselen-2 prevented radiation-induced mitochondrial injury and metabolic oxidative stress by targeting mitochondria, according to the findings. We previously reported that mitochondrial-derived ROS appears 3 h after irradiation, and excess ROS disrupt mitochondrial functions by degrading at a later stage in human fibroblasts [11]. Thus, melatonin and MitoEbselen-2 are effective at mitigating mitochondria-mediated oxidative stresses, even when administered after radiation.

Mitochondria are cellular damage sensors that secrete cytochrome C into the cytoplasm to cause cell death [8]. Many types of human tumors have been found to have mitochondrial DNA mutations, dysfunctions, and metabolic abnormalities [13,14]. We recently discovered that mitochondria play an important role in the formation of tumor microenvironment (niche) through fibroblast activation in radiation-related carcinogenesis [15]. Thus, it is believed that mitochondrial genomic instability and dysfunction play a role in radiation-related tumors [41,42]. Our present data indicated that melatonin and MitoEbselen-2 have been shown to prevent mitochondrial-mediated tumor microenvironment formation by suppressing the appearance of activated fibroblasts after radiation.

## 5. Conclusions

This study is the first to demonstrate that melatonin or MitoEbselen-2 suppressed mitochondrial radiation responses associated with cellular damage and radiation-related carcinogens. Both drugs have the potential to be a countermeasure agent against radiation’s acute and late effects. Pre-administration or post-administration of these drugs may have an effect in a variety of radiation scenarios, including expected or unexpected radiation exposure. For future recommendations, in vivo experiments will provide us with more solid evidence to apply to the clinical use of these agents.

## Figures and Tables

**Figure 1 genes-14-00045-f001:**
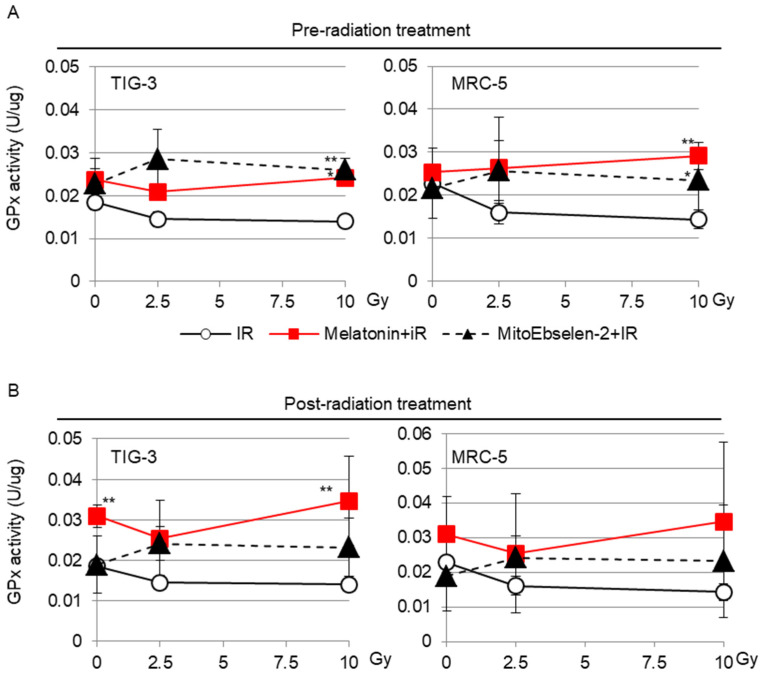
Effect of melatonin or MitoEbselen-2 on the GPx activity: the cells were treated with melatonin or MitoEbselen-2 before (pre-radiation treatment, (**A**)) or after (post-radiation treatment, (**B**)) IR. The GPx activity at 1 day after irradiation in indicated samples. Asterisk indicates a significant difference in the GPx activity of IR+ melatonin group and IR+ MitoEbselen-2 group compared with that of IR groups at same Gy level.

**Figure 2 genes-14-00045-f002:**
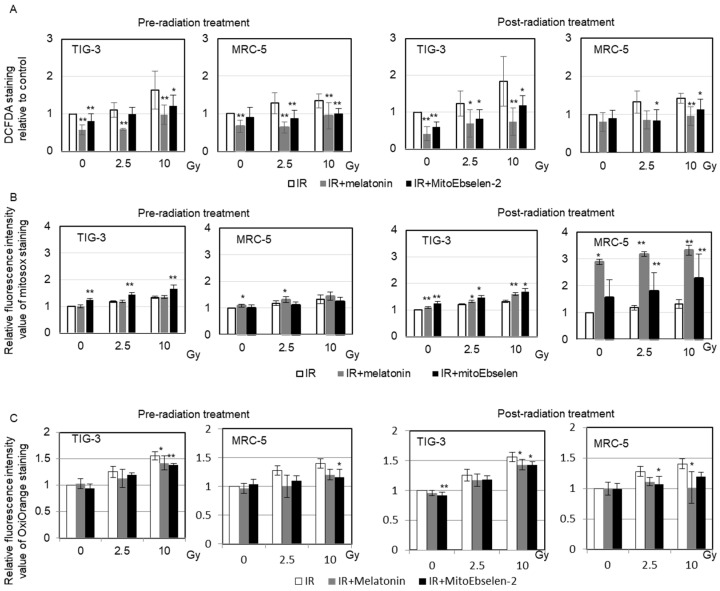
Measurements of the intracellular redox state, mitochondrial ROS and •OH. The relative fluorescence intensity values of the DCFDA (**A**), MitoSOX-red (**B**), and OxiORANGE (**C**) were normalized to non-irradiated controls. TIG-3 and MRC-5 cells were stained 1 day after X-ray exposure. Asterisk indicates a significant change in fluorescence intensity value of the staining in IR+ melatonin group and IR+ MitoEbselen-2 group compared with that of IR groups at same Gy level.

**Figure 3 genes-14-00045-f003:**
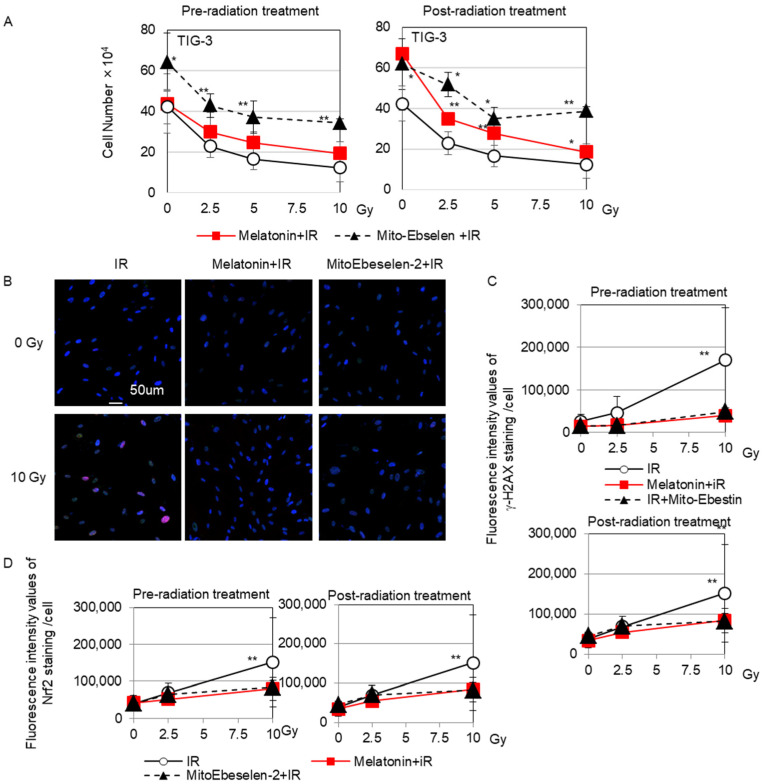
Cell growth, γ-H2AX foci formation, and nuclear accumulation of Nrf2 staining (**A**) The number of cells at 3 days after irradiation in indicated samples. Asterisk displays a significant difference in IR+ melatonin group and IR+ MitoEbselen-2 group compared with that of IR groups at same Gy level. (**B**) γ-H2AX foci (red) and nuclear accumulation of Nrf2 (green) were shown in irradiated TIG-3 cells. Scale bar = 50 µm. Fluorescence intensity of γ-H2AX (**C**) and Nrf2 (**D**) is shown in the graph. Asterisk displays a significant change in fluorescence intensity values of the staining compared with that of non-irradiated control cells.

**Figure 4 genes-14-00045-f004:**
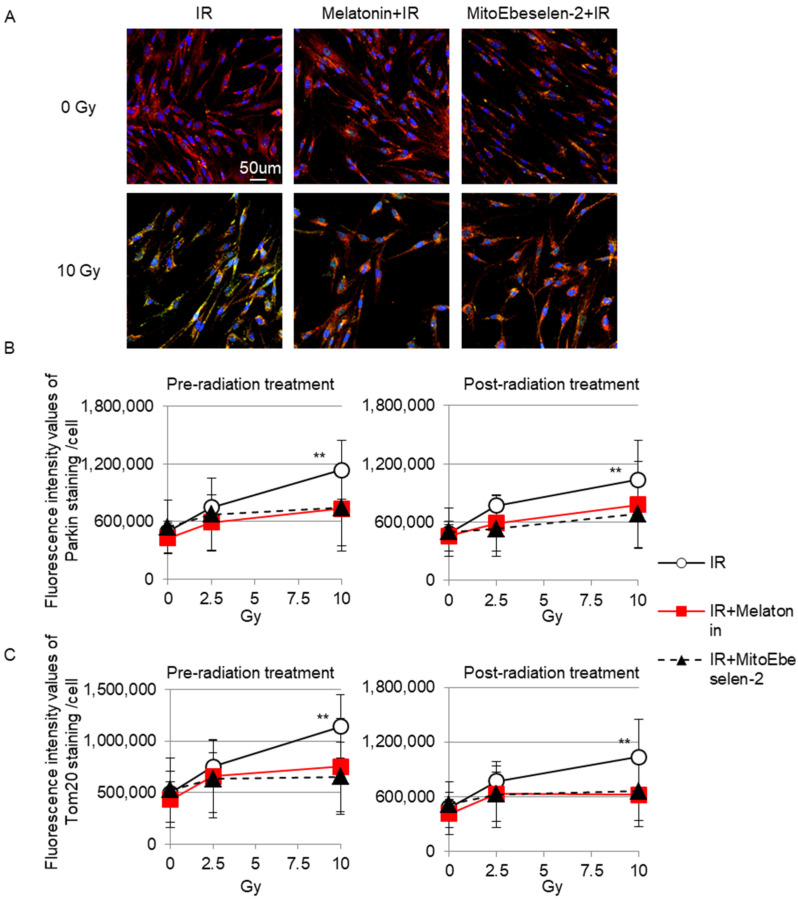
Mitochondrial radiation responses (**A**) Tom20 (red) and Parkin (green) positive staining were shown in irradiated TIG-3 cells. Scale bar = 50 µm. (**B**) Fluorescence intensity of Parkin (**B**) and Tom20 (**C**) staining is indicated in the graph. Asterisk displays a significant change in fluorescence intensity values of the staining compared with that of non-irradiated control cells.

**Figure 5 genes-14-00045-f005:**
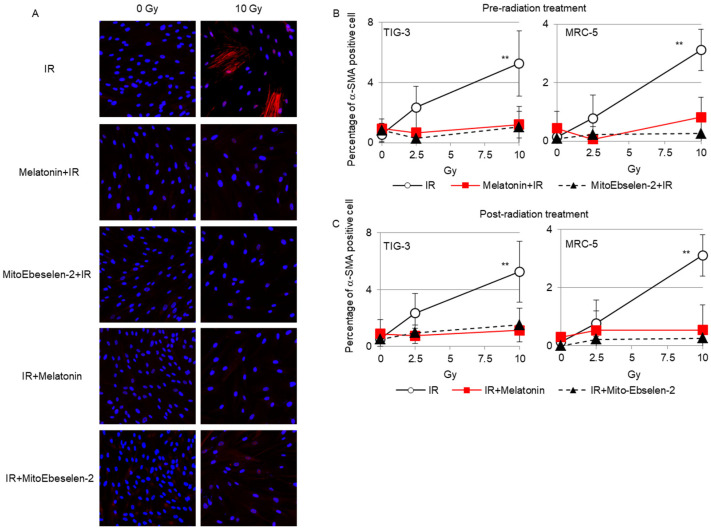
Induction of α-SMA-staining cells (**A**) α-SMA staining (red) was observed in irradiated TIG-3 cells. The ratio of α-SMA staining cells treated melatonin or MitoEbeselen-2 at pre-radiation (**B**) or post-radiation (**C**) is shown in the graph. Asterisk indicates a significant difference in the ratio of α-SMA-staining cells compared with that of non-irradiated control cells.

**Figure 6 genes-14-00045-f006:**
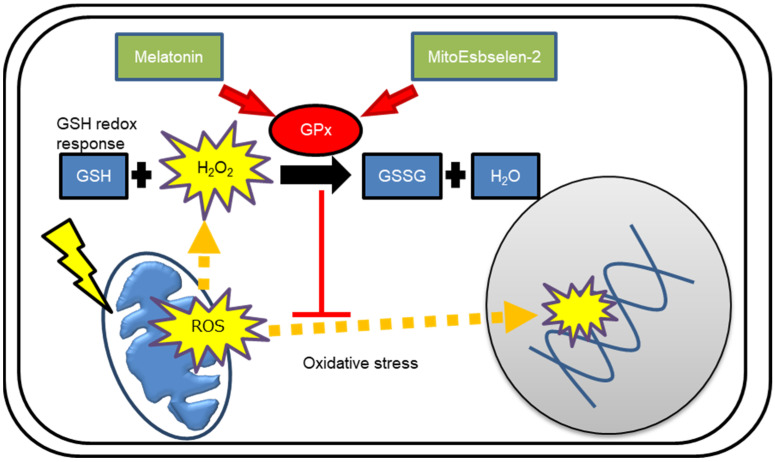
Schematic representation of role of melatonin or MitoEbselen-2 on GPx-mediated redox control.

## Data Availability

The data presented in this study are available on request from the corresponding author.

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
