# Peer review of "Melatonin and MitoEbselen-2 Are Radioprotective Agents to Mitochondria"

_genes, 2022, doi:10.3390/genes14010045_

Round 1

Reviewer 1 Report

This work is interesting and original, so I think it meets the requirements 

As a suggestion, authors should indicate in the first mention the meaning of abbreviations

Author Response

We appreciate editor and reviewers for spending their precious time. We have now carefully revised our manuscript according to the comments as indicated below. 

Referee: 1

This work is interesting and original, so I think it meets the requirements. As a suggestion, authors should indicate in the first mention the meaning of abbreviations.

Answer: Thank you for the reviewer’s comment. We spelled out the full term at its first mention.

Reviewer 2 Report

In this manuscript, Shimura et al. reported that mitochondrial injury and cell growth defects induced by radiation can be rescued or partially rescued by melatonin or MitoEbselen-2 treatment (pre- or post-administration). Further, they found that these drugs restored GPx activity, scavenged ROS, prevented activated fibroblasts after radiation. Though the study addresses an important area related to radiation injury and carcinogenesis, but the authors need to clarify their model, statistical analysis and experimental design. Moreover, Further experiments (i.e. confirmation of signaling pathways) are required and rationale is needed to delineate the model.

Below concerns (major and minor) should be address in a revised manuscript .

Major concerns:

  1. The innovation of this study.

As mentioned by the authors (Line 76-78), Reiter R.J. and Ianas O. have described that melatonin activated GPx and melatonin treatment protected a variety of tissues from induced free radical damage, possibly due to its ability to scavenger the -OH. In current study, the authors shown that “similar results were observed in fibroblast cells”. The innovation of this study is not good enough because previous researches have shown the similar results.

Some suggestions may enhance this study’s innovation mentioned below:

a. Paper needs to be put in better context with prior studies and current study in order to make readers easily distinguished what is already known, and what is truly novel about your study.

b. If this paper aim to develop a new radioprotective drug, animal experiment is good strategy. In vivo experiment will give us more information and provide more solid evidence to support authors’ point.

c. Base on given evidence, radiation inactivated GPx (Line 72), melatonin stimulated GPx and melatonin prevented radical damage (Line 77-78), a reasonable hypothesis came out. I do not insist on the following hypothesis but as a possible illustration of how the current results might be interpreted: GPx is master-regulator for radical damage and its activation status play important roles in preventing mitochondria injury and cell growth defect via maintenance of mitochondrial-derived ROS levels. Obviously, we need more molecular experiments (mutation of GPs, signaling pathways detection and so on) to support the hypothesis. The hypothesis I mentioned is just a possibility and it is more than welcome if you have other working model which can be supported by your data. A work model raised and confirmed by your work would be a shining point in your manuscript.

  1. Statistical analysis

For significant differences analysis, the comparison groups selected by the authors were very weird. For example, in Figure 1 upper left panel, authors tried to compare the effects of melatonin or MitoEbselen-2 pre-radiation treatment. The authors should choose IR group, IR+ melatonin group and IR+ MitoEbselen-2 group at same Gy level (for instance, 2.5Gy), then do statistical analysis to find is there any significant differences between these three groups, rather than every group comparing with IR+Gy0 group. Similar results can be found from Figure 1 to Figure 5, all these statistical analyses need to be redo.

  1. Some molecular experiments were needed.

Western blot for γ-H2AX, Nrf2 and α-SMA were needed as complementary data to support author’s point.

Minor concerns:

  1. Figures organization.

More detailed panel labels are required to make the figures more organized and easily to be quoted. In Figure 1, upper panel should be labeled as panel “A”, and lower panel should be labeled as panel “B”. In Figure 3B, the fluorescence result should be labeled as panel “B”, fluorescence intensity of γ-H2AX should be labeled as panel “C” and fluorescence intensity of Nrf2 should be labeled as panel “D”. Similar organizations are also needed for Figure 4 and Figure 5.

Author Response

Point-by-point answers to the comments:

We appreciate editor and reviewers for spending their precious time. We have now carefully revised our manuscript according to the comments as indicated below. 

Referee: 2

In this manuscript, Shimura et al. reported that mitochondrial injury and cell growth defects induced by radiation can be rescued or partially rescued by melatonin or MitoEbselen-2 treatment (pre- or post-administration). Further, they found that these drugs restored GPx activity, scavenged ROS, prevented activated fibroblasts after radiation. Though the study addresses an important area related to radiation injury and carcinogenesis, but the authors need to clarify their model, statistical analysis and experimental design. Moreover, further experiments (i.e. confirmation of signaling pathways) are required and rationale is needed to delineate the model.

Below concerns (major and minor) should be address in a revised manuscript.

Major concerns:

The innovation of this study.

As mentioned by the authors (Line 76-78), Reiter R.J. and Ianas O. have described that melatonin activated GPx and melatonin treatment protected a variety of tissues from induced free radical damage, possibly due to its ability to scavenger the -OH. In current study, the authors shown that “similar results were observed in fibroblast cells”. The innovation of this study is not good enough because previous researches have shown the similar results.

Answer: Mitochondria play a key role in the radiation effects through entire life. Radioprotective agents that targeted mitochondria are not fully understood. This study is first to demonstrate that melatonin or MitoEbselen-2 suppressed mitochondrial radiation responses associated with cellular damage and radiation-related carcinogens. Both drugs have the potential to protect against both acute radiation injury and late effects of radiation-related carcinogenesis in a variety of radiation scenarios, including expected or unexpected radiation exposure.

Some suggestions may enhance this study’s innovation mentioned below:

  1. Paper needs to be put in better context with prior studies and current study in order to make readers easily distinguished what is already known, and what is truly novel about your study.

Answer: Thank the reviewer for valuable comment. We have modified the manuscript followed by the reviewer’s suggestion.

  1. If this paper aims to develop a new radioprotective drug, animal experiment is good strategy. In vivo experiment will give us more information and provide more solid evidence to support authors’ point.

Answer: We agree with the reviewer’s comment about animal experiment. We described this as a future study. For the welfare of animals used in science, replacement (methods which avoid or replace the use of animals in research) is recommended. Therefore, we first carried out in vitro study to collect reliable scientific evidences. It will take a while to get approval for the animal experiment by Institutional Animal Ethic Committee.

  1. Base on given evidence, radiation inactivated GPx (Line 72), melatonin stimulated GPx and melatonin prevented radical damage (Line 77-78), a reasonable hypothesis came out. I do not insist on the following hypothesis but as a possible illustration of how the current results might be interpreted: GPx is master-regulator for radical damage and its activation status play important roles in preventing mitochondria injury and cell growth defect via maintenance of mitochondrial-derived ROS levels. Obviously, we need more molecular experiments (mutation of GPs, signaling pathways detection and so on) to support the hypothesis. The hypothesis I mentioned is just a possibility and it is more than welcome if you have other working model which can be supported by your data. A work model raised and confirmed by your work would be a shining point in your manuscript.

Answer: Thank the reviewer for giving us the idea for a reasonable hypothesis. A work model raised by this work is depicted in figure 6 in the revised manuscript. Radiation activates ataxia-telangiectasia mutated (ATM)-mediated mitochondrial respiration, which is required for energy supply to respond to nuclear DNA damage. Stimulating mitochondrial respiration increases the generation of delayed mitochondrial reactive oxygen species (ROS). Radiation inactivates the GPx and perturbs GSH-mediated redox control, causing metabolic oxidative stress and prolonged cell injury. In this study, we used melatonin and MitoEsbselen-2 to stimulate the GPx. GSH detoxifies ROS by converting them to GSH disulfide (GSSG) and H2O via the enzyme activity of GPx.

Statistical analysis

For significant differences analysis, the comparison groups selected by the authors were very weird. For example, in Figure 1 upper left panel, authors tried to compare the effects of melatonin or MitoEbselen-2 pre-radiation treatment. The authors should choose IR group, IR+ melatonin group and IR+ MitoEbselen-2 group at same Gy level (for instance, 2.5Gy), then do statistical analysis to find is there any significant differences between these three groups, rather than every group comparing with IR+Gy0 group. Similar results can be found from Figure 1 to Figure 5, all these statistical analyses need to be redo.

Answer: Followed by the reviewer’s suggestion, we compared among IR group, IR+ melatonin group and IR+ MitoEbselen-2 group at same Gy level in Figure 1, 2, 3A. In order to confirm whether melatonin or MitoEbselen-2 can prevent radiation responses of indicated markers, irradiated groups were compared with non-irradiated control cells in Figure 3B, 4, 5.

Some molecular experiments were needed.

Western blot for γ-H2AX, Nrf2 and α-SMA were needed as complementary data to support author’s point.

Answer: Western blotting analysis for α-SMA was previously reported (Cell Cycle 2020, 19, 3375-3385). The result indicated that radiation induced increase in α-SMA protein expression in human fibroblasts. However, western blotting cannot identify association between a-SMA expression and change in cellular morphology. Immunostaining revealed that activated fibroblast showed altered morphology (flatter and larger) with a fiber-like staining pattern of α-SMA (Ref. No15). Immunostaining with γ-H2AX and Nrf2 can also detect its focus formation and its association with nuclear accumulation of Nrf2 in irradiated cells.

Minor concerns:

Figures organization.

More detailed panel labels are required to make the figures more organized and easily to be quoted. In Figure 1, upper panel should be labeled as panel “A”, and lower panel should be labeled as panel “B”. In Figure 3B, the fluorescence result should be labeled as panel “B”, fluorescence intensity of γ-H2AX should be labeled as panel “C” and fluorescence intensity of Nrf2 should be labeled as panel “D”. Similar organizations are also needed for Figure 4 and Figure 5.

Answer: We corrected figures in the revised manuscript.

Reviewer 3 Report

This article used the glutathione peroxidase activator N-acetyl-5-methoxy-tryptamine (melatonin) and a mitochondria-targeted mimic of glutathione peroxidase MitoEbselen-2 to stimulate the glutathione peroxidase. However, there are few questions which must be addressed.

The abstract is well presented but methods are not clearly mentioned, which methods are used and what specific results were achieved. The abstract should be revise from line 17-25.

The article title has two main compounds “Melatonin and MitoEbselen-2” which act as radioprotective agent. Therefore, it should be discussing in introduction, like its composition, its mechanism of action, its literature review, its significance and why the author used these two as there are several other radioprotective elements exhibit. It can be adjusting in last paragraph of introduction onward line 75. Some of the required information is present in the following articles can be study and cited.

https://doi.org/10.3390/genes13101699, https://doi.org/10.3390/life12111922,

use section numbers. Like 2.1, 2.2

Add complete methods of GPx assay, and Cell counting,

Adjust figures to their respective sections according to journal format.

Add conclusion heading and modify the conclusion by adding future recommendations and impacts of the current study.

Author Response

We appreciate editor and reviewers for spending their precious time. We have now carefully revised our manuscript according to the comments as indicated below. 

Referee: 3

This article used the glutathione peroxidase activator N-acetyl-5-methoxy-tryptamine (melatonin) and a mitochondria-targeted mimic of glutathione peroxidase MitoEbselen-2 to stimulate the glutathione peroxidase. However, there are few questions which must be addressed.

The abstract is well presented but methods are not clearly mentioned, which methods are used and what specific results were achieved. The abstract should be revise from line 17-25.

Answer: We modified abstract as followed by the reviewer’s suggestion. A GPx activity assay kit was used to measure the GPx activity. ROS levels were measured by using some ROS indicators. Protein expression associated with the response of mitochondria to radiation was assessed using immunostaining.

The article title has two main compounds “Melatonin and MitoEbselen-2” which act as radioprotective agent. Therefore, it should be discussing in introduction, like its composition, its mechanism of action, its literature review, its significance and why the author used these two as there are several other radioprotective elements exhibit. It can be adjusting in last paragraph of introduction onward line 75. Some of the required information is present in the following articles can be study and cited.

https://doi.org/10.3390/genes13101699, https://doi.org/10.3390/life12111922,

Answer: Thanks the reviewer’s valuable comments. We described more about melatonin in introduction sections. Melatonin is present in almost all organisms to regulate the antioxidant system and circadian rhythms. Melatonin is tried to clinically use for sleep disorders of circadian etiology and neurological degenerative diseases. We focused on GPx-related agents, such as melatonin and MitoEsbselen-2, to develop radiation protective agents with fewer side effects.

use section numbers. Like 2.1, 2.2

Answer: We added section numbers in the revised manuscript.

Add complete methods of GPx assay, and Cell counting,

Answer: We described methods for GPx assay, and Cell counting as follows. The activities of GPx determined by evaluating the reduction of NADP+ to NADPH in the cell lysates following the instruction from the manufacturer’s protocols. NADPH was determined by measuring spectrophotometric absorbance at 340nm. Protein content in the enzymatic extracts was determined by the Bradford protein assay. The activity was expressed as μmol NADPH/min per mg protein.

The cells were incubated overnight, X-rayed the next day and then further incubated for 3 days. Cells were trypsinized and total number of cells was determined by using a hemocytometer and an optical microscope.

Adjust figures to their respective sections according to journal format.

Answer: We corrected.

Add conclusion heading and modify the conclusion by adding future recommendations and impacts of the current study.

Answer: We corrected

Round 2

Reviewer 2 Report

The paper is much improved after revision. I will be happy to accept this manuscript after polish the language.